# Evaluation of the Audicor Acoustic Cardiography Device as a Diagnostic Tool in Horses with Mitral or Aortic Valve Insufficiency

**DOI:** 10.3390/ani14020331

**Published:** 2024-01-21

**Authors:** Isabelle L. Piotrowski, Hannah K. Junge, Colin C. Schwarzwald

**Affiliations:** Clinic for Equine Internal Medicine, Vetsuisse Faculty, University of Zurich, 8057 Zurich, Switzerland; hjunge@vetclinics.uzh.ch (H.K.J.); cschwarzwald@vetclinics.uzh.ch (C.C.S.)

**Keywords:** horse, heart murmurs, heart sounds, phonocardiogram, cardiovascular

## Abstract

**Simple Summary:**

Cardiac murmurs caused by aortic or mitral insufficiency in horses often prompt further diagnostics to determine if the valvular insufficiencies affect the performance, safety, and life expectancy of the horse. Ambulatory acoustic cardiography (Audicor^®^) provides qualitative and quantitative information on hemodynamics and cardiac mechanical function and may aid in clinical diagnosis and prognostication of heart disease. The objective of this study was to investigate the use of acoustic cardiography in horses with aortic or mitral valve insufficiency. The results do not suggest any added clinical value of using the Audicor^®^ device to detect heart murmurs and quantify alterations in cardiac hemodynamics and mechanical function in horses with aortic and mitral insufficiency that are not experiencing heart failure.

**Abstract:**

Mitral and aortic valve insufficiencies have been commonly reported in horses. The objective of this study was to establish the use of acoustic cardiography (Audicor^®^) in horses with aortic (AI) or mitral valve insufficiency (MI). A total of 17 healthy horses, 18 horses with AI, and 28 horses with MI were prospectively included. None of the horses was in heart failure. Echocardiography and Audicor^®^ analyses were conducted. Electromechanical activating time (EMAT), rate-corrected EMATc, left ventricular systolic time (LVST), rate-corrected LVSTc, and intensity and persistence of the third and fourth heart sound (S3, S4) were reported by Audicor^®^. Graphical analysis of the three-dimensional (3D) phonocardiogram served to visually detect murmurs. Audicor^®^ snapshot variables were compared between groups using one-way ANOVA followed by Tukey’s multiple-comparisons test. The association between Audicor^®^ snapshot variables and the corresponding echocardiographic variables was investigated by linear regression and Bland–Altman analyses. Heart murmurs were not displayed on Audicor^®^ phonocardiograms. No significant differences were found between Audicor^®^ variables obtained in clinically healthy horses and horses with valvular insufficiency. The Audicor^®^ device is unable to detect heart murmurs in horses. Audicor^®^ variables representing cardiac function are not markedly altered, and their association with corresponding echocardiographic variables is poor in horses with valvular insufficiency that are not in heart failure.

## 1. Introduction

Mitral and aortic valve insufficiencies have been reported in up to 65% of horses, with older horses being more commonly affected than younger horses [1,2,3,4,5]. Assessing the clinical relevance of cardiac auscultatory findings usually requires further diagnostics to determine whether the valvular insufficiency affects the performance, safety, and life expectancy of the horse [1,6,7]. Ancillary techniques such as transthoracic echocardiography, resting and exercising electrocardiography (ECG), noninvasive blood pressure measurement, and cardiac catheterization require appropriate equipment, advanced training, and experience. They are usually only performed by specialists in the field or in larger referral centers [8]. Ambulatory acoustic cardiography (Audicor^®^) allows comprehensive, noninvasive, operator-independent, continuous assessment of the electrical and hemodynamic characteristics of the heart in people with cardiac disease [9]. The technology uses a phonocardiogram to characterize heart sounds by their acoustic components (i.e., timing, frequency, and intensity). In conjunction with a simultaneously recorded electrocardiogram, it provides qualitative and quantitative information on hemodynamics and cardiac mechanical function and may aid in clinical diagnosis and prognostication of heart disease [10]. Audicor^®^ has not been validated for murmur detection in people or other species. However, the visual representation of the phonocardiogram offers the potential to visually identify heart murmurs. In people with mitral or aortic valve insufficiency, a strong correlation has been found between the presence of a third heart sound (S3) and abnormal cardiac hemodynamics [11,12]. In addition, increased intensity and persistence of S3 are associated with larger left atrial and left ventricular dimensions, lower ejection fraction, left ventricular diastolic dysfunction, and restrictive filling patterns [13,14,15]. In healthy horses, all four heart sounds can frequently be heard on auscultation [2,16,17]. Although S3 can reach a higher intensity in horses with heart disease, accurate methods of quantifying the strength and persistence of S3 are lacking, and the diagnostic and prognostic value currently remains uncertain. The Audicor^®^ technology could overcome this shortcoming. Recently, the usability of the Audicor^®^ device was examined, reference intervals were calculated, and the reproducibility of the analyses was evaluated in a study in healthy horses [18]. Data on its use in horses with valvular disease are lacking to date. The aim of this study was to examine the use of a second-generation Audicor^®^ acoustic cardiography monitor in horses with aortic (AI) or mitral insufficiency (MI). The first hypothesis was that Audicor^®^ acoustic cardiography would be able to detect diastolic and systolic heart murmurs. The second hypothesis was that Audicor^®^ variables representing diastolic and systolic cardiac function, including the intensity of S3, would be altered in horses with valvular insufficiency.

## 2. Materials and Methods

### 2.1. Study Sample

Horses with AI or MI presented to the Vetsuisse Faculty of the University of Zurich between 2019 and 2021 were prospectively enrolled in the study. All horses had to be older than 1 year and taller than 148 cm (withers height), while there were no restrictions regarding sex, breed, training status, or athletic discipline. All horses had to be free of dietary supplements or medications containing adrenergic or anticholinergic agents, vasodilators, diuretics, ACE inhibitors, cardiac glycosides, or antiarrhythmics for 1 month prior to the start of the study. In addition, no analgesics or anti-inflammatory drugs were allowed 1 week prior to Audicor^®^ recordings and echocardiography. Healthy control horses were presented to the hospital for reasons other than cardiovascular disease. They were deemed healthy based on medical history, physical examination, cardiac and pulmonary auscultation, echocardiography, and electrocardiography. The study was approved by the district veterinary office of the Canton of Zurich. Owner consent was obtained for each animal prior to its enrollment into the study.

### 2.2. Clinical and Echocardiographic Examinations

On presentation, horses were assessed by clinical examination. Cardiac murmurs were classified according to point of maximal intensity (PMI), timing within the cardiac cycle, character, and intensity on a scale of 1–6/6 [19]. Transthoracic echocardiographic examinations, including 2-dimensional (2D), M-mode, color flow Doppler, and tissue Doppler methods, were performed by one of four experienced operators according to a standardized protocol. The quality of the recordings, validity, and consistency of the results were confirmed for all cases by the senior cardiology expert. Both the operators and the senior cardiologist were blinded to the Audicor^®^ results. Cardiac structures, valvular competence, chamber dimensions, left atrial (LA) mechanical function, and left ventricular (LV) systolic and diastolic function were assessed using right parasternal long-axis and short-axis views [20,21,22,23,24,25,26]. All examinations were performed in a quiet setting with the unsedated horse being restrained by an experienced handler. A high-end echocardiograph (GE Vivid e95 Ultrasound system; GE Healthcare, Glattbrugg, Switzerland) with a phased array 4D transducer (4Vc-D; GE Healthcare) operated in 2D mode at a frequency of 1.9/4.0 MHz (octave harmonics) was used. A base-apex ECG was recorded simultaneously for timing purposes. Digital raw data were stored as still images or cine loops for subsequent offline analysis (EchoPAC; GE Healthcare). Three representative, nonconsecutive cardiac cycles were measured and averaged for each variable. The heart rate (HR) of a measured cycle was calculated based on the RR interval (in milliseconds) preceding the analyzed cycle (HR = 60,000/RR). The echocardiographic variables used in this study are summarized in Appendix A. Briefly, 2D and M-mode echocardiographic variables used to assess LA size and function included maximum left atrial diameter measured at end-systole (LAD_max_), maximum left atrial area measured at end-systole (LAA_max_), maximum left atrial short-axis area measured at end-systole (LA_sx_A_max_), and active left atrial fractional area change (active LA FAC). Variables used to assess LV size and function included the internal diameter of the LV at end-diastole (LVID_d_) and peak systole (LVID_s_), LAD_max_/LVID_d_ ratio, left ventricular internal volume at end-diastole (LVIV_d_) and peak systole (LVIV_s_), left ventricular fractional shortening (LV FS), and left ventricular ejection fraction (LV EF). Tissue Doppler imaging (TDI)-derived variables characterizing LA and LV mechanical function included the pre-ejection period (PEP_m_), ejection time (ET_m_), PEP_m_/ET_m_ ratio, index of myocardial performance (IMP_m_), early-diastolic LV wall motion velocity during the phase of rapid ventricular filling (E_m_), late-diastolic LV wall motion velocity at the time of atrial contraction (A_m_), and E_m_/A_m_ ratio. Because the study sample contained horses with a wide range of body weights (BWT), the dimensional measurements were corrected for differences in weight according to the principles of allometric scaling [27,28]. Specifically, the measurements of chamber dimensions were normalized to a BWT of 500 kg using the following equations:Chamber diameter (500) = Measured chamber diameter/BWT^1/3^ × 500^1/3^
Chamber area (500) = Measured chamber area/BWT^2/3^ × 500^2/3^
Chamber volume (500) = Measured chamber volume/BWT × 500

### 2.3. Assessment of Severity of Valvular Insufficiency

The severity of AI and MI was categorized as mild, moderate, or severe, based on a scoring system considering the size of the regurgitant jet relative to the receiving chamber as well as the size and shape of the LV and the LA. Aortic regurgitation was categorized using a scoring system that was based on a previously described score [29] modified according to the institution’s standards by using LV volume estimates and defining the left ventricular outflow tract as the receiving chamber (Appendix A). Mitral regurgitation was scored using a similar scoring system that had been established at the authors’ institution (Appendix A). All horses were scored independently by two operators (CS and HJ), who were unaware of any previous classifications for each horse. Any scoring discrepancies between the operators were resolved by a second simultaneous review by the two operators. Additional insufficiencies, i.e., tricuspid and pulmonic insufficiencies, were graded based on the number of imaging planes in which the high-velocity jet could be observed, the duration of the regurgitant signal, and the area of the high-velocity jet relative to the receiving chamber [1].

### 2.4. Categorization of Horses

The primary focus of the study was on horses with AI or MI. The horses were grouped according to the primarily affected valve, as judged by the clinician conducting the physical examination and the echocardiogram, considering the murmurs heard on auscultation and the severity of valvular insufficiency judged by echocardiography. Horses with additional valvular insufficiency (pulmonary and/or tricuspid valve insufficiency) were included in the study if the insufficiency was considered mild or of lesser severity.

### 2.5. Audicor^®^ Acoustic Cardiography Recordings

The Audicor^®^ device was installed after clinical and echocardiographic examination. Data recordings were obtained by a single operator (IP) with the second-generation Audicor^®^ Dx Patch device (ApoDx Technologies, Taipei, Taiwan), as described in a previous study [18] (Appendix A). The battery-operated device is equipped with two dry electrodes for recording a digital surface ECG signal and a high-fidelity accelerometer for heart sound recordings. The internal digital storage capabilities allow up to 24 h of data to be recorded at a sampling rate of 500 Hertz (Hz). The recordings were performed in a standardized manner, starting between 05:00 and 08:00 pm and ending between 05:00 and 08:00 am the next day. After completion of the recordings, the raw data files were transferred via a USB connection from the Dx Patch device to a laptop computer for later processing and analyses.

### 2.6. Audicor^®^ Data Processing and Analyses

The Audicor^®^ software algorithm relies on proprietary wavelet-based signal processing techniques and time–frequency analyses of the raw data. Data processing and analyses were performed by a single operator (IP) using specialized data analysis software (CA 300, Inovise Medical Inc., Beaverton, OR, USA). Manual correction of the ECG was subsequently performed by a single operator, as the automated wave detection is based on human complex morphology, occasionally misinterpreting the equine T and P waves as a QRS complex. Missed or erroneously marked beats were corrected. Artifact detection was based on an automated algorithm combined with visual control by the operator.

For the purpose of this study, five consecutive, good-quality 10-second snapshots analyses were performed from each overnight recording, based on the raw data recorded between 8.00 and 9.00 pm. If the recording quality was insufficient for analysis, the following best 5 consecutive snapshots were taken. From the 5 consecutive snapshots (50 s in total), the 10-second segment with the best quality was selected to create a three-dimensional (3D) representation of the phonocardiogram, showing the four heart sounds (S1–S4) characterized by timing, sound frequency, and acoustic energy. Visual inspection of the 3D illustration of the phonocardiogram to detect heart murmurs was performed by a single operator (IP) blinded to the groups. The variables generated by the snapshot analysis were the following: heart rate (HR), electromechanical activation time (EMAT), heart-rate-corrected EMAT (EMATc), left ventricular systolic time (LVST), heart-rate-corrected LVST (LVSTc), strength (as a function of intensity and persistence) of S3, strength of S4, and systolic dysfunction index (SDI, a function of QRS duration, QT interval, EMATc, and S3) [18]. A glossary of all variables is provided in Appendix A. Accounting for the lower heart rate of horses compared to people and the proprietary Audicor^®^ analysis algorithm, and based on a previous study showing an enhanced reproducibility of Audicor^®^ results when combining five consecutive snapshots [30] the median values of EMAT, EMATc, LVST, LVSTc, and SDI and the maximum values of the power of S3 and S4 were extracted from the five snapshots for further analyses.

### 2.7. Statistical Analyses

All statistical and graphical analyses were performed with commercially available software (Microsoft Excel 365; GraphPad PRISM for Windows, v9.0.0). Horses were assigned to one of three groups: healthy control horses (HC), horses with aortic insufficiency (AI), and horses with mitral insufficiency (MI). Descriptive statistics were calculated, and normally distributed data were expressed as mean and standard deviation (SD). To detect differences between the groups (HC, AI, MI) in heart rate, echocardiographic variables, and Audicor^®^ snapshot variables, a one-way analysis of variance (ANOVA) with Tukey’s multiple-comparisons test was performed. Homogeneity of variances was assessed with a graphical display of the data, and validity of the normality assumption was confirmed by assessment of histograms and normal probability plots of the residuals. Bonferroni correction was performed to adjust for the family-wise error rate within a set of variables describing LA and LV size and function. The association between Audicor^®^ snapshot variables (i.e., EMAT, LVST, EMAT/LVST) and the corresponding echocardiographic variables (i.e., PEP_m_, ET_m_, PEP_m_/ET_m_) was investigated by linear regression analyses. The appropriateness of the linear model was assessed with the graphical display of the data and the residuals. The *p* values and the coefficient of determination (R^2^) were reported. Bland–Altman analyses were performed, and mean biases and 95% limits of agreement were calculated for Audicor^®^ snapshot variables and the corresponding echocardiographic variables. The level of significance was set at *p* < 0.05.

## 3. Results

Audicor^®^ acoustic cardiography recordings could be obtained for all 63 horses included in the study. A description of the study sample is given in Appendix A. Briefly, a total of 17 horses were included in the HC group. The group consisted of 15 Warmbloods and 2 Standardbreds, 8 of which were mares and 9 of which were geldings. They had a body weight of 535 ± 33 kg and were between 5 and 18 years old (12 ± 4.6 years (mean ± SD)). The MI group contained 28 horses (26 Warmbloods, 1 Arabian, 1 Andalusian; 13 mares, 12 geldings, 3 stallions). They weighed 554 ± 68 kg and were between 4 and 28 years old (13.6 ± 6.4 years). The MI was graded as mild in 19 horses, moderate in 9 horses, and severe in none of them. The AI group contained 18 horses (13 Warmbloods, 3 Arabians, 2 Friesians; 6 mares, 11 geldings and 1 stallion). They weighed 554 ± 77 kg and were between 10 and 27 years old (19 ± 4.8 years). The AI was graded as mild in 3 horses, moderate in 9 horses, and severe in 6 horses.

The age differed significantly between groups (F-test, *p* = 0.0004). Horses with AI were significantly older than horses with MI (mean difference 5.6 years; 95% confidence interval of difference of means 1.6 to 9.6 years) and the HC group (mean difference 7.4 years; 95% CI 2.9 to 11.8 years), while the age was not significantly different in the MI vs. the HC group (mean difference 1.8 years; 95% CI -2.3 to 5.9 years). Body weight was not significantly different between groups (F-test, *p* = 0.5202).

Table 1 summarizes the comparison of HR and echocardiographic variables of cardiac size and function between the three groups. The HR did not differ between groups. The LAD_max_ (500) and LAA_max_ (500) were higher in the MI group and the AI group, respectively, compared to the HC group, while no difference was detected between the MI and the AI group. The LA_sx_A_max_ (500) was higher in the AI group compared to the HC group only. The active LA FAC was not significantly different between groups. Both the LVID_d_ (500) and the LVIV_d_ (500) were higher in the AI group compared to the HC and the MI groups, respectively; LVIV_d_ (500) was higher in the MI group compared to the HC group. The LAD_max/_LVIDd ratio was significantly lower in the AI group compared to the other two groups, while it was not significantly different between MI and HC groups. The active LV FS was not significantly different between groups. The LV EF was significantly lower in the MI group compared to the HC group, while it did not differ between the AI group and the other groups.

Among the TDI-derived variables of LA and LV mechanical function, PEP_m_ was significantly prolonged in the MI group compared to the HC group, while it did not differ between the AI group and the other groups. The AI group had a significantly longer ET_m_ than the HC group and MI group. PEP_m/_ET_m_ was significantly higher in the MI group than the AI group and the HC group. No significant differences in IMP_m_, E_m_, A_m_, or E_m_/A_m_ were found between groups. After Bonferroni correction to adjust for the family-wise error rate among echocardiographic variables, only changes in LAD_max_ (500), LAA_max_ (500), LVID_d_ (500), LVIV_d_ (500), LAD_max/_LVID_d_, and ET_m_, remained statistically significant.

Based on visual inspection of the selected 10-second segment of the 3D phonocardiograms and the printed display of the sound trace, none of the murmurs heard on auscultation could be identified in any of the horses (Figure 1).

Table 2 summarizes the comparison of the Audicor^®^ variables obtained in clinically healthy horses and horses with valvular insufficiency. No significant differences between groups were found. Neither S3 strength nor S4 strength were significantly different between groups.

Linear regression analyses did not indicate significant relationships between PEP_m_ and EMAT, ET_m_ and LVST, or PEP_m/_ET_m_ and EMAT/LVST. Bland–Altman analysis indicated large biases and wide limits of agreement between the respective corresponding variables (Appendix A).

## 4. Discussion

Audicor^®^ acoustic cardiography reference intervals have previously been published in healthy large-breed horses [18]. This is the first study aiming at detecting heart murmurs using Audicor^®^ acoustic cardiography and comparing acoustic cardiography variables between healthy horses and horses with AI and MI. The echocardiographic characterization of the study sample indicated that horses in the MI group had proportional LA and LV enlargement, whereas horses in the AI group primarily had LV enlargement. Among the systolic time intervals measured in this study, AI was primarily associated with the prolongation of ET_m_. This corresponds well to the overall findings in other studies [20] and relates to the pathophysiology of disease [21]. The hypothesis that Audicor^®^ acoustic cardiography could aid in detecting systolic and diastolic heart murmurs was not confirmed by the results of this study. Based on the graphical analysis of the sound traces and 3D phonocardiograms, it was not possible to detect or classify any of the heart murmurs present in the study sample. However, 13/28 horses with MI had a systolic murmur grade ≥ 4/6 on auscultation, 13/18 horses with AI had a diastolic murmur grade ≥ 4/6, and 4 horses in the study sample even had a murmur grade 6/6, suggesting that the murmurs should have been sufficiently loud to be detected. The failure of murmur detection was most likely due to the fact that the device could not be placed at the point of maximum intensity of the murmurs but rather had to be placed underneath a surcingle that was fastened in the girth position. Furthermore, there may have been motion artifacts negatively impacting the sound recordings. Finally, the frequency range of the microphone and the filtering algorithms of the Audicor^®^ device are specifically designed to identify heart sounds in people and detect murmurs with sound components under 200 Hz [31]. In horses, the frequency of murmurs, particularly musical murmurs, may exceed 200 Hz [32], calling into question the clinical value of the Audicor^®^ technology in detecting and characterizing heart murmurs in horses.

Also, no association was found between the presence of valvular insufficiency and the intensity of S3 and S4. Acoustic cardiography objectively detects S3, and in humans, an S3 strength of ≥5 indicates its presence. In people over 40 years of age, S3 is considered pathological and is associated with increased LV filling pressure and impaired LV contractility [33]. The presence of S4 (i.e., S4 strength ≥5) is also pathological and associated with LV stiffness and increased LV end-diastolic pressure [34]. In horses, S3 and S4 are generally considered physiologic, and they can be commonly heard on auscultation even in the absence of heart disease [19]. In our study sample, 13/63 horses had S3 or S4, as detected by auscultation (Appendix A). However, the strength of S3 and S4 detected by Audicor^®^ was low, and there was no statistically significant difference between groups. Again, not being able to place the device at the point of maximum intensity of heart sounds could be one reason for the low strengths of S3 and S4 recorded. Other possible reasons could be the infeasibility of extrapolating values from the human scale to the horse scale and the lack of sufficiently severe heart disease in the study sample.

In humans with heart disease, the detection of left ventricular systolic dysfunction and impaired myocardial contractility is clinically important, particularly for monitoring therapy response and the prediction of prognosis in patients with heart failure [35]. The LV EF is one of the most commonly used indices for identifying LV systolic dysfunction. However, the LV EF is often imprecise; it lacks the sensitivity required to detect subtle LV dysfunction, and its measurement is time-consuming [36]. Studies have shown that EMAT measured by Audicor^®^ is prolonged in people with impaired LV function and that it allows accurate detection of LV dysfunction [9,35,37,38]. In patients with heart failure, an abnormal EMAT is strongly correlated with impaired LV contractility and hence a lower LV EF, end-systolic elastance, and peak isovolumetric LV pressure at the end-diastolic volume. In addition, an abnormal EMAT is also associated with a higher end-systolic volume index, end-diastolic volume index, and dyssynchrony [39]. The detection of early subclinical LV dysfunction in horses using Audicor^®^ technology could provide useful additional information, for example, in the evaluation of horses with murmurs, in the assessment of horses with poor performance, or as a prognostic indicator. However, in this study, no differences in Audicor^®^ variables between groups were detected. Direct statistical comparison of EMAT and LVST with PEP_m_ and ET_m_ did not indicate a statistically significant association and a relatively large bias with wide limits of agreement. Considering that 9/28 horses were graded with moderate MI and 15/18 horses were graded with moderate or severe AI, likely associated with some degree of hemodynamic alterations and structural remodeling, these results do not support the assumption that Audicor^®^-derived variables are more sensitive than echocardiography in detecting disease-related alterations in cardiac hemodynamics and function in horses. The clinical value of the second-generation Audicor^®^ technology for detecting hemodynamic alterations and LV dysfunction in horses with AI and MI therefore seems limited.

Some limitations must be considered when interpreting the results of this study. The horses in the HC group were not case-matched to the MI and AI groups; the AI group was significantly older than the HC group. Most of the horses had additional (although minor) valvular insufficiencies, and therefore it was not possible to assess hemodynamic alterations in pure MI and AI. The study sample did not include horses with severe MI or horses with obvious signs of heart failure. Therefore, the hemodynamic and functional alterations might simply have been too subtle to be detected by the Audicor^®^ technology. Also, the group size did not allow for more detailed subgroup analyses and comprehensive assessment of the diagnostic value Audicor^®^ variables in horses with varying severity of AI and MI. However, the study sample was drawn from the average patient population of a large referring hospital with high expertise in equine cardiology, which likely sees patients with severe cardiac disease more frequently than non-specialized clinics or field practitioners. Furthermore, as mentioned above, the position of the Audicor^®^ sound sensor could not be optimized for the varying points of maximum intensity of the heart sounds and heart murmurs in individual horses. Moreover, Audicor^®^ analyses are based on algorithms designed for people. Finally, Audicor^®^ recordings and echocardiographic examinations were not conducted simultaneously, partly explaining the lack of association and large bias between corresponding variables.

## 5. Conclusions

In conclusion, the results of this study do not suggest added clinical value with the use of the ambulatory second-generation Audicor^®^ device to detect heart murmurs and to quantify subtle alterations in cardiac hemodynamics and mechanical function in horses with AI and MI that are not in heart failure. Future studies will be conducted using the next-generation Audicor^®^ device, addressing some of the technical shortcomings of the current device, such as sensor placement relative to the point of maximum intensity of heart sounds and heart murmurs.

## Figures and Tables

**Figure 1 animals-14-00331-f001:**
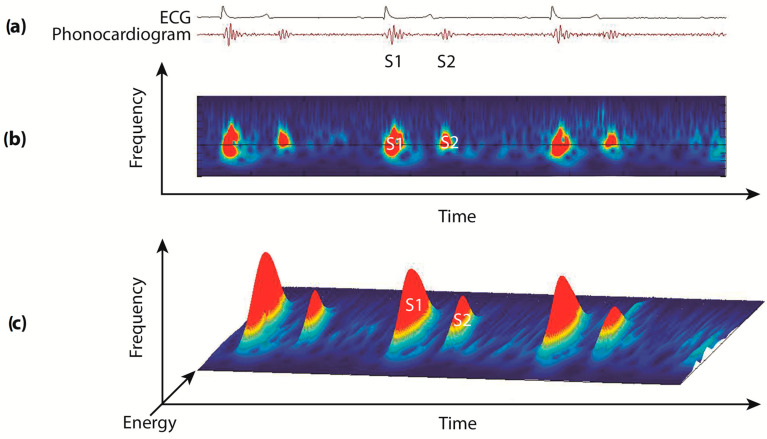
Electrocardiogram, phonocardiogram, and two-dimensional (2D) and three-dimensional (3D) phonocardiogram recorded by Audicor^®^. (**a**) Electrocardiogram and phonocardiogram recorded by Audicor^®^ in a horse with moderate AI and a grade 4/6 diastolic murmur heard on auscultation. The first (S1) and second (S2) heart sounds are shown on the phonocardiogram, while the diastolic murmur is not visible; (**b**) the two-dimensional representation of the phonocardiogram shows the sound frequency on the *y*-axis, while the acoustic energy is color-coded (low-energy signals in blue, intermediate-energy signals in yellow, high-energy signals in red). The diastolic murmur is not visible; (**c**) three-dimensional representation of the phonocardiogram with two heart sounds (S1 and S2) characterized by timing, sound frequency, and acoustic energy. The diastolic murmur is not visible.

**Table 1 animals-14-00331-t001:** Comparison of echocardiographic variables of cardiac function between the groups with valvular insufficiency and the healthy control group.

Variable	Unit	Healthy Control (HC) *n* = 17 Mean ± SD	Mitral Insufficiency (MI) *n* = 28 Mean ± SD Mean Difference (95% CI of Difference of Means) ^a^	Aortic Insufficiency (AI) *n* = 18 Mean ± SD Mean Difference (95% CI of Difference of Means) ^b,c^	*p* Value (F-Test)
Heart rate	min^−1^	37 ± 6	37 ± 8	34 ± 5	0.4973
			1 (−4 to 6), *p* = 0.9298	−2 (−7 to 4), *p* = 0.7481	
				−2 (−7 to 3), *p* = 0.4669	
LAD_max_ (500)	cm	11.4 ± 0.7	12.2 ± 0.7	12.2 ± 0.9	0.0017 *
			0.8 (0.2 to 1.3), *p* = 0.0042	0.9 (0.2 to 1.5), *p* = 0.0042	
				0.1 (−0.5 to 0.7), *p* = 0.9392	
LAA_max_ (500)	cm^2^	86 ± 8	97 ± 11	95 ± 10	0.0021 *
			11 (4 to 18), *p* = 0.0019	9 (1 to 17), *p* = 0.0216	
				−2 (−8 to 5), *p* = 0.8282	
LA_sx_A_max_ (500)	cm^2^	100 ± 8	108 ± 11	111 ± 15	0.0217
			8 (−1 to 17), *p* = 0.0701	11 (1 to 21), *p* = 0.0227	
				3 (−6 to 11), *p* = 0.7334	
Active LA FAC	%	24 ± 5	24 ± 8	28 ± 9	0.1983
			1 (−5 to 6), *p* = 0.9699	4 (−2 to 10), *p* = 0.2434	
				4 (−2 to 9), *p* = 0.2627	
LVID_d_ (500)	cm	10.8 ± 0.8	11.7 ± 0.7	12.8 ± 1.9	0.0001 *
			0.9 (0 to 1.7), *p* = 0.0511	2.0 (1.1 to 3.0), *p* = 0.0001	
				1.1 (0.3 to 2.0), *p* = 0.0060	
LVIV_d_ (500)	mL	1001 ± 144	1221 ± 187	1488 ± 379	0.0001 *
			220 (36 to 404), *p* = 0.0152	488 (286 to 690), *p* = 0.0001	
				268 (87 to 448), *p* = 0.0021	
LAD_max_/LVID_d_	-	1.06 ± 0.07	1.05 ± 0.07	0.95 ± 0.13	0.0010 *
			−0.01 (−0.08 to 0.06), *p* = 0.9103	−0.11 (−0.18 to −0.03), *p* = 0.0028	
				−0.09 (−0.16 to −0.03), *p* = 0.0028	
LV FS	%	39 ± 4	39 ± 7	41 ± 5	0.4194
			0 (−5 to 4), *p* = 0.9740	2 (−3 to 7), *p* = 0.6071	
				2 (−2 to 7), *p* = 0.4045	
LV EF	%	73 ± 3	70 ± 5	73 ± 5	0.0140
			−4 (−7 to −1), *p* = 0.0224	−1 (−4 to 3), *p* = 0.8913	
				3 (0 to 6), *p* = 0.0688	
PEP_m_	msec	118 ± 28	138 ± 16	126 ± 22	0.0170
			20 (3 to 36), *p* = 0.0145	8 (−9 to 26), *p* = 0.5021	
				−11 (−27 to 5), *p* = 0.2161	
ET_m_	msec	427 ± 17	420 ± 38	453 ± 21	0.0015 *
			−7 (−29 to 15), *p* = 0.7123	26 (2 to 50), *p* = 0.0277	
				34 (12 to 55), *p* = 0.0012	
PEP_m_/ET_m_	-	0.277 ± 0.067	0.331 ± 0.054	0.280 ± 0.056	0.0039
			0.054 (0.010 to 0.097), *p* = 0.0118	0.003 (−0.045 to 0.050), *p* = 0.9883	
				−0.050 (−0.094 to −0.008), *p* = 0.0159	
IMP_m_	-	0.311 ± 0.043	0.345 ± 0.095	0.310 ± 0.079	0.2642
			0.033 (−0.026 to 0.092), *p* = 0.3759	−0.001 (−0.067 to 0.064), *p* = 0.9986	
				−0.035 (−0.094 to −0.025), *p* = 0.3471	
E_m_	cm/sec	31 ± 4	33 ± 4	31 ± 7	0.4005
			2 (−2 to 5), *p* = 0.4525	0 (−4 to 4), *p* = 0.9852	
				−2 (−5 to 2), *p* = 0.5510	
A_m_	cm/sec	11 ± 3	14 ± 5	13 ± 5	0.0884
			3 (0 to 6), *p* = 0.0714	2 (−2 to 6), *p* = 0.3915	
				−1 (−4 to 2), *p* = 0.6963	
E_m_/A_m_	-	3.0 ± 1.1	2.6 ± 1.0	2.6 ± 0.9	0.3199
			−0.4 (−1.2 to 0.3), *p* = 0.3176	−0.4 (−1.2 to 0.4), *p* = 0.4647	
				0.1 (−0.7 to 0.8), *p* = 0.9869	

^a^ MI group vs. HC group; ^b^ AI group vs. HC group; ^c^ AI group vs. MI group. * Significant after Bonferroni correction for family-wise error rate among the echocardiographic variables (*p* < 0.0031).

**Table 2 animals-14-00331-t002:** Summary statistics of Audicor^®^ variables obtained in clinically healthy horses and horses with valvular insufficiency.

Variable	Unit	Healthy Control (HC) *n* = 17 Mean ± SD	Mitral Insufficiency (MI) *n* = 28 Mean ± SD Mean Difference (95% CI of Difference of Means) ^a^	Aortic Insufficiency (AI) *n* = 18 Mean ± SD Mean Difference (95% CI of Difference of Means) ^b,c^	*p* Value (F-Test)
Heart rate	min^−1^	35 ± 5	37 ± 6	37 ± 8	0.5225
			2 (−3 to 7), *p* = 0.5749	2 (−3 to 7), *p* = 0.5676	
				0 (−4 to 5), *p* = 0.9927	
EMAT	msec	104 ± 20	107 ± 21	95 ± 21	0.1350
			3 (−12 to 19), *p* = 0.8598	−9 (−26 to 8), *p* = 0.3870	
				−13 (−28 to 2), *p* = 0.1169	
EMATc	%	6 ± 1	6 ± 2	5 ± 2	0.5078
			0 (−1 to 2), *p* = 0.8032	0 (−2 to 1), *p* = 0.8922	
				−1 (−2 to 1), *p* = 0.4858	
LVST	msec	506 ± 33	488 ± 33	513 ± 43	0.0547
			−19 (−46 to 8), *p* = 0.2225	7 (−23 to 36), *p* = 0.8479	
				26 (−1 to 52), *p* = 0.0607	
LVSTc	%	29 ± 4	29 ± 4	31 ± 5	0.2115
			0 (−4 to 3), *p* = 0.9722	2 (−2 to 6), *p* = 0.3829	
				2 (−1 to 6), *p* = 0.2053	
EMAT/LVST	-	0.208 ± 0.048	0.225 ± 0.052	0.188 ± 0.056	0.0733
			0.017 (−0.021 to 0.059), *p* = 0.5290	−0.019 (−0.062 to 0.023), *p* = 0.5203	
				−0.037 (−0.075 to 0.001), *p* = 0.0596	
S3 strength	-	5 ± 1	5 ± 1	5 ± 1	0.4264
			0 (−1 to 1), *p* = 0.4589	0 (−1 to 1), *p* = 0.9710	
				0 (−1 to 1), *p* = 0.6042	
S4 strength	-	4 ± 1	4 ± 1	5 ± 1	0.0654
			0 (−1 to 1), *p* = 0.8643	1 (0 to 2), *p* = 0.0767	
				1 (0 to 1), *p* = 0.1322	

^a^ MI group vs. HC group; ^b^ AI group vs. HC group; ^c^ AI group vs. MI group.

## Data Availability

Data are contained within the article and supplementary materials.

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
