# Peer review of "Evaluation of the Audicor Acoustic Cardiography Device as a Diagnostic Tool in Horses with Mitral or Aortic Valve Insufficiency"

_animals, 2024, doi:10.3390/ani14020331_

Round 1

Reviewer 1 Report

Comments and Suggestions for Authors

The Audicor device is used in humans to help to detect cardiac failure. However, detection of murmurs can maybe possible but is there a study that proofs validation of the Audicor device for murmur detection in humans/others? This is not well described in the introduction part.

In the “Introduction” some parts are misleading. It gives impression that most information about heart sounds and cardiac hemodynamics are gathered in a simple way such as with an “Audicor” device but if you check references then information was gathered for example via more invasive cardiac catheterization.

The Audicor device is more valuable in humans >40y with cardiac failure. What about horses?

Line 48-50: is this really examiner independent?

Reviewer 2 Report

Comments and Suggestions for Authors

Overall, the manuscript is well written. I have just a few comments on notes in an effort to improve the quality and clarity of the manuscript.

It appears that the control animals were not case matched with the horses with AI or MI. This is an error in study design given the number of variables that may exist in clinical cases, effort should be made to match animals by breed, age, weight etc. This limitation should either be rectified or included as a study limitation in the discussion.

What was the interobserver variability in M-mode and doppler analysis (referenced in line 94)? Were these investigators blinded.

The fact that horses may have had multiple valvular insufficiencies (line 153) is a confounding limitation that should be included in the limitations, and this information should be included in the demographic information.

An explanation the manual ECG correlation (line 172) should be included.

It Is not clear why a considerable amount of the data relevant to this study has been put in supplemental material rather than organized in a way that is in the published manuscript.

Provide a table with the demographic information for the control, MI and AI horses (derived from S5).

Modify S6 and S7 so they can be published in the manuscript and provide appropriate table descriptions.

Reword the results and discussion with reference to these changes.

Minor:

Line 16: cut word ambulatory from the sentence.

Line 41-47: Reword this sentence and watch for grammar errors in line 47

Line 63: What is the basis for this statement (“The Audicor technology could..”) please explain and provide a reference supporting that assertion.

Line 68-71: Reword this odd compound sentence and watch for grammar errors in line 70-71.

References are not listed consistency (ex. 25, 26) these should be standardized and meet journal criteria.   

Comments on the Quality of English Language

See notes above on grammar.  I recommend an additional round of proof reading and copy editing to clean up small English issues.

Reviewer 3 Report

Comments and Suggestions for Authors

In the present work the authors report the results of the application on horses with aortic and mitral insufficiency of a device that simultaneously records the electrocardiogram and the phonocardiogram (AUDICOR), providing more precise information on cardiac activity compared to the application of the individual methods.

Although the authors conclude that this device does not add significant information in horses with valvula insufficiences compared to traditional listening, ECG and echocardiography, I believe that the work can be published in Animals. Currently, the application of precision medicine in the veterinary field is being discussed, transferring devices designed for humans to different animal species, and it is important to understand if these methods are really applicable and if they add precision to the diagnosis and prognosis of different pathologies. In my opinion the work is well designed from an experimental point of view and well written in English. I am not qualified to evaluate statistic methods. 

I suggest  the authors to add for a better reader's  understanding a picture of a horse included in the study with the Audicor device applied.

​

Round 2

Reviewer 1 Report

Comments and Suggestions for Authors

The quality of the manuscript is improved. There is only a minor remark that should be changed in the introduction part. Evolution goes fast! Cfr line 50: Transthoracic echocardiography, resting and exercising electrocardiography, non-invasive blood pressure measurement and, rarely, cardiac catheterization are not only performed in larger referral centers. All is now also done in an ambulant way, in other words, all these exams are also performed on location (at home of the horse owner, riders,…).